# Genetic potential for aerobic respiration and denitrification in globally distributed respiratory endosymbionts

Daan R. Speth [1,4] ✉, Linus M. Zeller[1], Jon S. Graf[1], Will A. Overholt [2], Kirsten Küsel [2,3] & Jana Milucka [1] ✉

The endosymbiont *Candidatus* Azoamicus ciliaticola was proposed to generate ATP for its eukaryotic host, an anaerobic ciliate of the Plagiopylea class, fulfilling a function analogous to mitochondria in other eukaryotic cells. The discovery of this respiratory endosymbiosis has major implications for both evolutionary history and ecology of microbial eukaryotes. However, with only a single species described, knowledge of its environmental distribution and diversity is limited. Here we report four complete, circular metagenome assembled genomes (cMAGs) representing respiratory endosymbionts inhabiting groundwater in California, Ohio, and Germany. These cMAGs form two lineages comprising a monophyletic clade within the uncharacterized gammaproteobacterial order UBA6186, enabling evolutionary analysis of their key protein complexes. Strikingly, all four cMAGs encode a cytochrome $cbb_3$ oxidase, which indicates that these endosymbionts have the capacity for aerobic respiration. Accordingly, we detect these respiratory endosymbionts in diverse habitats worldwide, thus further expanding the ecological scope of this respiratory symbiosis.

Host beneficial endosymbionts (HBEs) are widespread in eukaryotes, and often evolve from a parasitic or pathogenic ancestor[1]. The best studied HBEs are nutritional symbionts of insects that provide their hosts with essential molecules (such as vitamins or cofactors), which the hosts cannot synthesize or obtain from their diet[2,3]. Another example are defensive endosymbionts that synthesize compounds that protect the host or its offspring against predators or pathogens[4,5]. Beyond insect hosts, intracellular endosymbionts also perform a wide range of beneficial functions in protists, such as photosynthesis, scavenging of hydrogen or other metabolic products, or complementing catabolic pathways (reviewed in ref.[6]). The recently discovered protist endosymbiont *Ca*. A. ciliaticola fulfills yet another function, using a denitrifying respiratory chain to generate ATP that can be supplied to its ciliate host[7]. The Plagiopylean ciliate host of *Ca*. A. ciliaticola was proposed to harbor only metabolically reduced organelles known as mitochondrion-related organelles (MROs)[8,9], possibly in the form of hydrogenosomes. Thus the host is likely incapable of (aerobic) respiration, although it might still be able to generate ATP in the MROs, or through substrate level phosphorylation in the cytoplasm[7,10]. The endosymbiont *Ca*. A. ciliaticola also lacks the capability for aerobic respiration, restricting the ecological niche of its ciliate host to permanently anoxic habitats, such as the hypolimnion of a meromictic lake. Interestingly, virtually all organisms capable of denitrification are facultative anaerobes[11,12], and therefore it was hypothesized that *Ca*. A. ciliaticola evolved from a predecessor that was capable of both aerobic respiration and denitrification[7]. To investigate this possibility, we searched for genomes of endosymbionts related to *Ca*. A. ciliaticola in publicly available environmental metagenomic sequencing datasets.

[1]Department of Biogeochemistry, Max Planck Institute for Marine Microbiology, Bremen, Germany. [2]Aquatic Geomicrobiology, Friedrich Schiller University, Jena, Germany. [3]Cluster of Excellence Balance of the Microverse, Friedrich Schiller University, Jena, Germany. [4]Present address: Division of Microbial Ecology, Centre for Microbiology and Environmental Systems Science, University of Vienna, Vienna, Austria. ✉e-mail: daan.speth@univie.ac.at; jmilucka@mpi-bremen.de

In this work we describe the discovery for four additional closed genomes representing respiratory endosymbionts. The five genomes form a monophyletic clade and have strongly conserved gene content and genomic features, indicating a shared function in their host. Using the expanded genomic coverage, we confidently place this endosymbiont clade in the UBA6186 order, which we accordingly propose to rename to *Azoamicales*. We show that the four newly retrieved genomes encode the capability for aerobic respiration in addition to respiration of nitrogen oxides, and that the genes required for both were acquired horizontally. Finally, we use the expanded genome coverage to show that the respiratory endosymbionts are globally distributed.

## Results

### Discovery and genomic features of novel endosymbiont genomes

We used the *tlcA* gene, which encodes for an ATP/ADP transporter that is crucial for the proposed ATP supplying function of *Ca*. A. ciliaticola, to identify genomes of putative respiratory endosymbionts. Using this strategy, we recovered one complete circular cMAG originating from groundwater samples taken in California[13] and assembled three complete circular cMAGs from groundwater samples of a carbonate-rock-fracture aquifer taken in Germany (1 cMAG)[14,15] and groundwater samples from Ohio (2 cMAGs)[16]. All four cMAGs share genomic features characteristic of obligate endosymbionts, including extreme genome reduction (284–373 kb genome size), low GC content (20–26 %) and a high protein coding density (91–92 %) (Fig. 1, Supplementary Table 1). Additionally, all four cMAGs contain a complete respiratory denitrification pathway (Fig. 1, Supplementary Data 2), but have undergone extensive loss of genes for the biosynthesis of vitamins, amino acids, and other essential metabolites (Supplementary Data 3). These genomic similarities strongly suggest that, like *Ca*. A ciliaticola, these cMAGs represent obligate respiratory endosymbionts that perform denitrification.

The four cMAGs (hereafter: genomes) form a monophyletic clade with *Ca*. A. ciliaticola, within the UBA6186 order of the Gammaproteobacteria (Fig. 1, Supplementary Fig. 1, Supplementary Notes). The groundwater genomes belong to two distinct clades, with the two Ohio genomes forming a lineage with the lacustrine *Ca*. A. ciliaticola, and the California and Germany genomes forming a novel lineage (Fig. 1). Based on the pairwise average amino acid

identity (AAI) and average nucleotide identity (ANI) values[17,18] between the five genomes (Supplementary Data 4) we propose that they represent five species grouped in two genera within the *Candidatus* Azoamicaceae family (hereafter: *Azoamicaceae*). The two genera are the previously established *Candidatus* Azoamicus (hereafter: *Azoamicus*) and a novel genus, for which we propose the name *Candidatus* Azosocius (hereafter: *Azosocius*; see Supplementary Notes for discussion of species and genera delineation). We propose the names *Candidatus* Azoamicus viridis and *Candidatus* Azoamicus soli for the two Ohio genomes and *Candidatus* Azosocius agrarius and *Candidatus* Azosocius aquiferis for the California and Germany genomes, respectively. *Candidatus* Azosocius agrarius is the type species of the *Azosocius* genus.

The genomes of the *Azosocius* species are nearly identical in size (352 and 353 kpb for *Ca*. A. agrarius and *Ca*. A. aquiferis, respectively) and they are larger than those of *Ca*. A. soli (284 kpb) and *Ca*. A. ciliaticola (293 kbp); however, the largest genome belonged to *Ca*. A. viridis (374 kbp). A comparative genomic analysis of the five *Azoamicaceae* genomes revealed that their collective protein coding gene content amounts to only 470 genes (Fig. 2a), of which 254 are shared between all five genomes (68–87 % of the protein coding gene complement of individual genomes). This conserved set includes the genes for an electron transport chain and ATP synthase (Complexes I, II, III, and V), a complete denitrification pathway for reduction of nitrate to dinitrogen gas, ATP/ADP transporter, iron sulfur cluster biosynthesis, molybdopterin cofactor biosynthesis, and cellular information processing (Fig. 2d, Supplementary Data 2). 43 additional genes are present in multiple (2-4) genomes across both *Azoamicus* and *Azosocius* lineages (Fig. 2a), indicating ongoing loss of genes after the lineages diverged. Pseudogene prediction revealed 3-8 putative pseudogenes in the *Azoamicaceae* genomes (Supplementary Data 5), providing further support for ongoing genome erosion. In addition to the genes shared between the two lineages, there are 107 genes unique to *Azoamicus* and 66 genes unique to *Azosocius* (Fig. 2b), as further discussed in the Supplementary Notes. Analysis of single nucleotide variants (SNVs) revealed only 0 − 65 SNVs in the groundwater *Azoamicaceae* genomes (Supplementary Data 6). However, this limited genomic variability may be a consequence of the low coverage of the *Azoamicaceae* in the source datasets, as the more deeply sequenced metagenomes of *Ca*. A. ciliaticola[7] showed higher variation (74 − 2856 SNVs). Nonetheless, these analyzes indicate that strain differentiation might exist in these

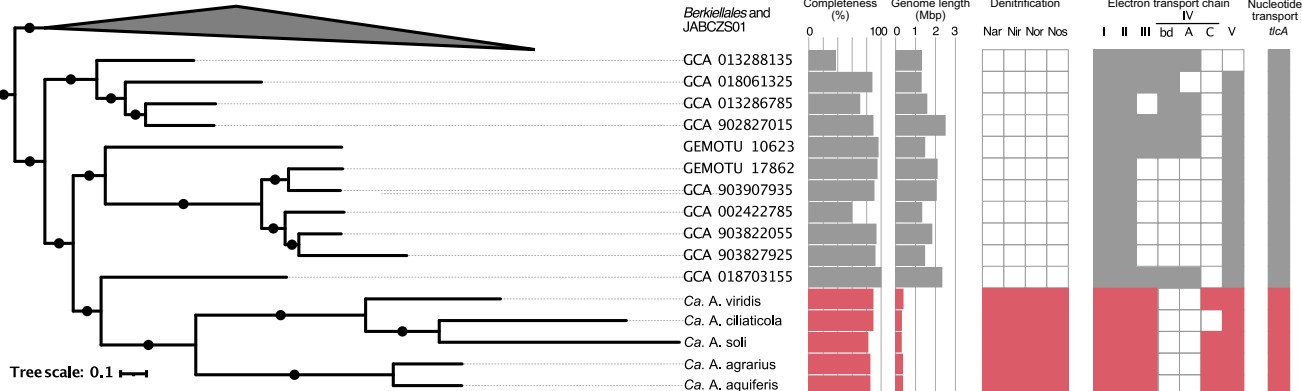

**Fig. 1 | Phylogeny and selected genomic features of the UBA6186 order.** Concatenated marker gene phylogeny of the gammaproteobacterial orders Berkiellales, JABCZS01 and UBA6186. The expanded section shows the UBA6186 order, including the *Azoamicaceae*. The tree annotations indicate, from left to right, MAG completeness as estimated by Anvi'o (in %), genome length (in Mbp), presence (filled squares) or absence (empty squares) of the denitrification pathway, the electron transport chain complexes NADH dehydrogenase (I), succinate dehydrogenase (II), *bc₁* complex (III), cytochrome *bd* oxidase (IV - bd), cytochrome *caa₃*

oxidase (IV - A), cytochrome *cbb₃* oxidase (IV - C), and ATP synthase (V), and presence of at least one copy of the *tlcA* gene encoding an ATP/ADP transporter. The genomic features of Azoamicaceae are indicated in red, while the features of the other UBA6186 genomes are shown in gray. Sequences were obtained from genomes included in the genome taxonomy database (GTDB) and genomic catalog of Earth's microbiomes (GEM) databases. Black circles on the branches indicate bootstrap values higher than 90%.

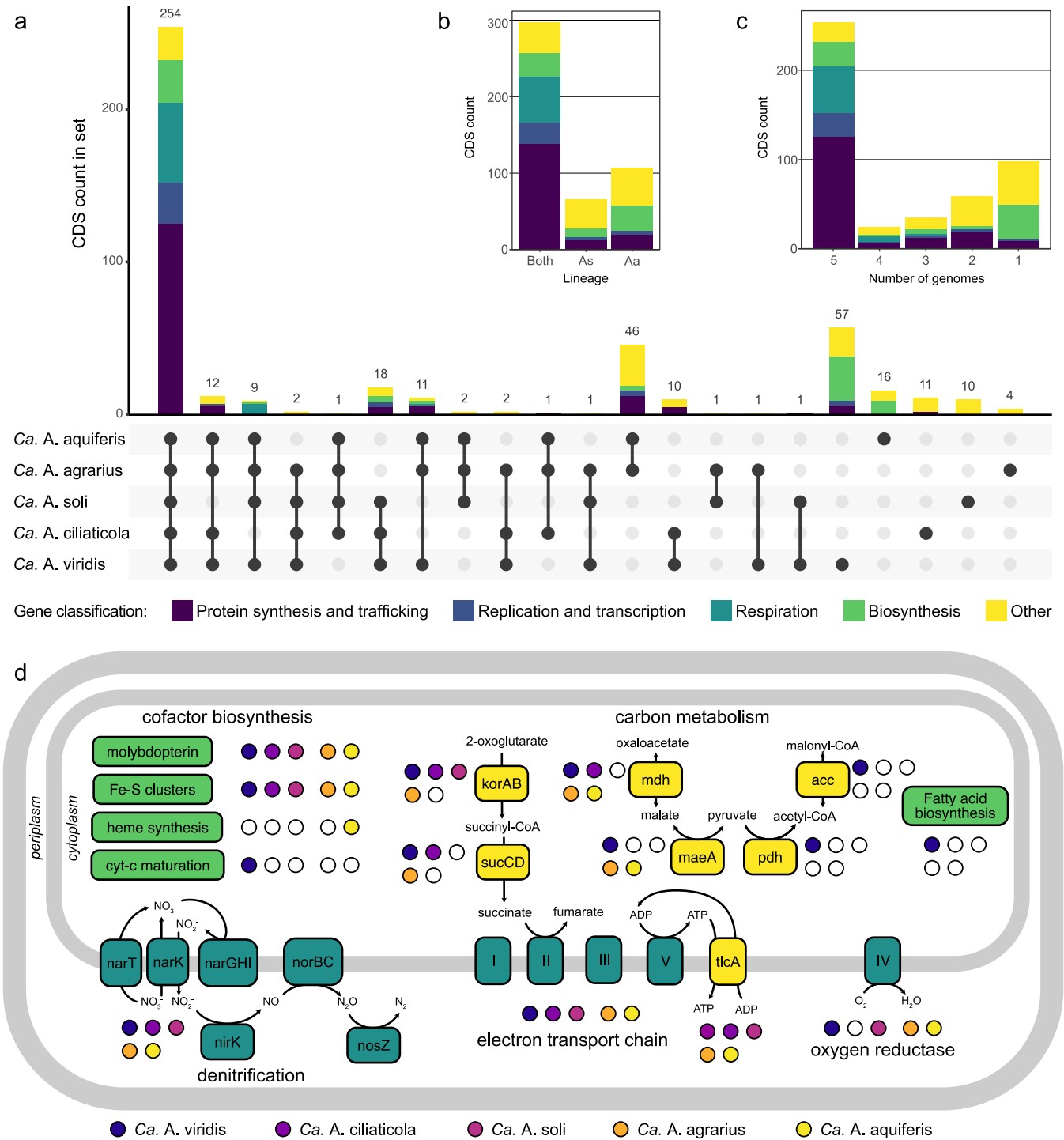

**Fig. 2 | Gene distribution and pathway overview in the Azoamicaceae genomes.** **a** UpSet plot showing the coding sequence (CDS) content shared between, or unique to, the individual *Azoamicaceae* genomes. Each vertical bar represents the number of genes in an intersecting set between the genomes. The filled circles connected by lines indicate which genomes contribute genes to the intersecting set. Sets are ordered by the number of genomes contributing, and then the number of genes in the set. Vertical bars are colored according to the broad classification of genes that can be found in Supplementary Data 2. Total coding sequence numbers for the genomes are *Ca.* A aquiferis 352; *Ca.* A agrarius 347; *Ca.* A soli 299; *Ca.* A ciliaticola 311; *Ca.* A viridis 378 (**b**, inset) Distribution of the gene content shown in panel A, summarized by genus. Aa: *Azoamicus*, As: *Azosocius*. (**c**, inset) Distribution of the CDS content shown in panel A summarized by the number of genomes containing each gene. (**d**) Key pathways in the *Azoamicaceae* genomes that could enable generation of ATP for the host cell from both denitrification and aerobic respiration. Boxes indicate protein complexes or pathways, with background shading corresponding to the categories in panels (**a–c**). Filled and empty circles indicate the presence/absence of a complex or pathway in the genome. Roman numerals in the boxes representing the electron transport chain complexes indicate: (I) NADH dehydrogenase, (II) succinate dehydrogenase, (III) $bc_1$ complex, (IV) cytochrome $cbb_3$ oxidase, (V) ATP synthase.

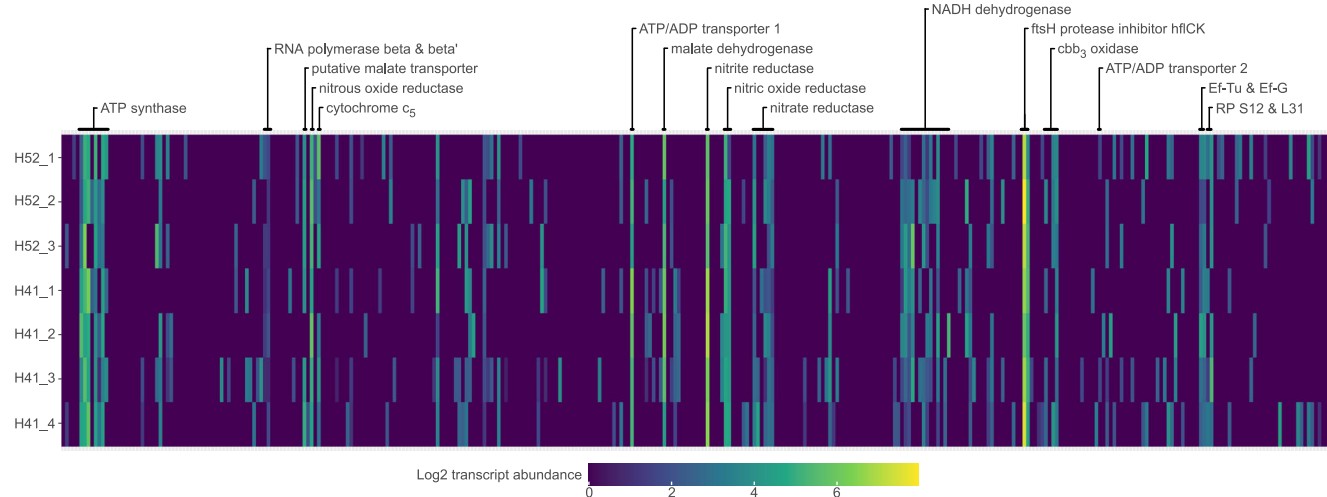

**Fig. 3 | In situ gene expression of *Ca*. A. aquiferis genes in groundwater metatranscriptomes from Hainich CZE.** Heatmap depicting the gene expression of 352 protein coding genes (columns) in the genome of *Ca*. A. aquiferis across seven metatranscriptome samples (rows) taken from anoxic groundwater of well H5-2 and oxic groundwater of well H4-1 at the Hainich critical zone exploratory (CZE) during two sampling campaigns. The genes are arranged in order according to their position in the *Ca*. A. aquiferis genome, with highly expressed genes and other selected genes annotated above the heatmap. The gene expression was normalized to "transcripts per thousand", and then log₂ transformed to highlight the genome-wide expression pattern. Ef-Tu: Elongation factor Tu; Ef-G: Elongation factor G; RP S12 & L31: Ribosomal proteins S12 and L31. Accession numbers of the datasets are available in the Method section.

microbial symbionts, with a so-far unknown effect on the symbiotic association with its host.

## Presence and transcription of high affinity cytochrome *c* oxidase in groundwater Azoamicaceae

Notably, among the genes conserved in the four genomes is an operon consisting of *ccoNO(Q)P*, encoding for a high affinity cytochrome *cbb₃* oxygen reductase (complex IV)[19]. The *cbb₃* operon is complete in three of the four genomes, only *Ca*. A. soli appears to have lost the *ccoQ* gene that encodes for the small non-catalytic subunit[20]. The cytochrome *cbb₃* oxygen reductases were initially thought to be restricted to Proteobacteria, but have since also been found in genomes of organisms from other bacterial phyla[21]. In organisms encoding a cytochrome *cbb₃* oxidase, small $c_4$ and $c_5$ cytochromes can form a branch point in the respiratory chain, allowing electrons from complex III to flow to either the *cbb₃* oxidase during aerobic respiration or a nitrite reductase during denitrification. All four *Azoamicaceae* genomes encode the diheme cytochrome $c_4$ that acts as the natural electron donor to the cytochome *cbb₃* oxidase in *Vibrio cholerae*[22] and *Neisseria meningitidis*[23], as well as the $c_5$ monoheme cytochrome that shuttles electrons from complex III to nitrite reductase in *Neisseria meningitidis*[23]. These small cytochromes are also present in *Ca*. A. ciliaticola, which lacks the *cbb₃* oxidase, strongly suggesting that *Ca*. A. ciliaticola has lost its terminal oxidase secondarily. This is further supported by the presence of a *cbb₃* oxidase in closely related *Ca*. A. viridis and *Ca*. A. soli genomes from the *Azoamicus* genus (Fig. 1).

We confirmed the transcription of the *ccoNOQP* operon in *Ca*. A aquiferis using data from 18 groundwater metatranscriptome datasets from the Hainich Critical Zone Exploratory (CZE)[24], from which the *Ca*. A aquiferis genome was assembled (Supplementary Data 7). Sufficient reads matching *Ca*. A. aquiferis genes for analysis of genome wide transcription patterns (327–1138 reads) could be recovered from seven of these datasets, which represent two wells sampled at two time points. The overall observed transcription pattern was very similar in all seven datasets (Fig. 3, Supplementary Data 7). As previously observed for *Ca*. A. ciliaticola, the *nirK*, *norB*, and *nosZ* genes were amongst the highest transcribed genes for *Ca*. A aquiferis in all groundwater samples, with transcription of the other genes of the denitrification pathway lower but still detectable[7]. The catalytic

subunit of the cytochrome *cbb₃* oxidase (encoded by *ccoN*) was also consistently highly transcribed, at a similar level as e.g. the *narG* gene (encoding for the catalytic subunit of nitrate reductase). The *ccoO* and *ccoP* genes encoding the other core subunits of the cytochrome *cbb₃* oxidase were also highly transcribed in some but not all datasets (Fig. 3, Supplementary Data 5). Interestingly, high *ccoNOP* gene transcription was detected in samples with markedly different dissolved oxygen concentrations[15]. Samples from well H5-2 originate from typically anoxic groundwater in low permeable marlstones and dissolved oxygen was not detected at the time of RNA sampling[25]. In contrast, groundwater samples from the permeable fracture aquifer accessed at well H4-1 contained ~6 mg/L dissolved oxygen at the time of RNA sampling[25], which is typical for this sampling site[15]. The comparably high levels of transcription of the cytochrome *cbb₃* oxidase operon in oxic and anoxic groundwater at both timepoints (Fig. 3) may indicate that the terminal oxidase is constitutively expressed in *Ca*. A. aquiferis, as previously observed for *cco1* of *Pseudomonas aeruginosa*[26].

The presence of a cytochrome *cbb₃* oxidase operon in *Azoamicaceae* symbionts suggests that like their free-living counterparts, symbiotic denitrifying bacteria also typically have the capacity to respire oxygen in addition to nitrogen oxides. However, oxygen respiration can be secondarily lost as demonstrated by the example of *Ca*. A. ciliaticola. In the case of *Azoamicaceae* symbionts, the presence and expression of a terminal oxidase raises the compelling possibility that *Azoamicaceae* respiratory endosymbionts respire oxygen, instead of, or in addition to, host mitochondria.

## Evolutionary history of respiratory enzymes in *Azoamicaceae*

Given the importance of nitrate- and oxygen-respiring genes for the function of the *Azoamicaceae* respiratory endosymbionts, we used phylogenetic analyzes to investigate the evolutionary history of those genes. We assume vertical inheritance when *Azoamicaceae* genes form a well-supported clade with their UBA6186 relatives, whereas genes that are absent from the entire UBA6186 group, are considered more likely to be horizontally acquired by the *Azoamicaceae*. All the core complexes of the electron transport chain (complex I, II, III and V) are monophyletic and appear to be vertically inherited from the UBA6186 clade (Fig. 1, Supplementary Fig. 2). In contrast, the cytochrome *cbb₃* oxidase (complex IV) appears to be horizontally transferred from the

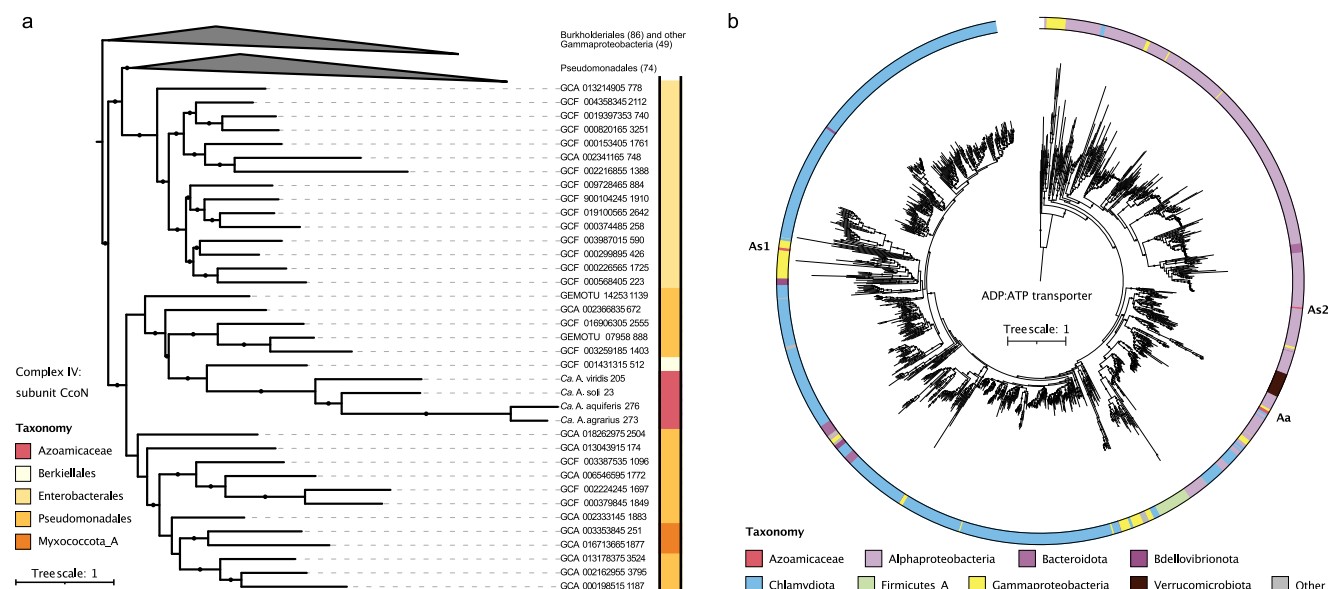

**Fig. 4 | Phylogeny of terminal oxidase and ATP/ADP translocase. a** Phylogenetic tree based on amino acid sequences of the catalytic subunit of the *cbb₃* cytochrome c oxidase (*ccoN*). Color strip indicates the GTDB assigned order within the class Gammaproteobacteria (or the phylum Myxococcota_A) of the respective genome containing the *ccoN* gene. Collapsed wedges contain gammaproteobacterial sequences primarily originating from Pseudomonales and Burkholderiales. **b** Phylogenetic tree based on amino acid sequences of the ATP/ADP transporter *tlcA* showing three independent origins for the Azoamicaceae genes. Both Azosocius genomes contain two copies (As1 & As2), and the three Azoamicus genomes contain third distinct *tlcA* copy (Aa). Color strip indicates the GTDB assigned taxonomy at the phylum or class (for Pseudomonadota) of the genome containing the tlcA gene. Sequences were obtained from genomes included in the genome taxonomy database and genomic catalog of Earth's microbiomes databases. Black circles on the branches indicate bootstrap values higher than 80% (**a**) or 90% (**b**).

*Pseudomonadales*, with the closest homologs present in *Alcanivoraceae* and *Berkiellales* (Fig. 4a). The horizontal acquisition of *cbb₃* oxidase by the *Azoamicaceae* is supported by the observation that none of the UBA6186 genomes, and only a single genome in the sister clade *Berkiellales*, contain genes for a cytochrome *cbb₃* oxidase. Instead, both clades typically encode a cytochrome *caa₃* oxidase and a cytochrome *bd*-type alternative oxidase, which are both absent from the five *Azoamicaceae* genomes (Fig. 1). The key genes encoding for the core enzymes of the denitrification pathway (*narG*, *nirK*, *norB* and *nosZ*) also appear to be horizontally acquired, from four different bacterial donor lineages (Supplementary Figs. 3–6). The four gene clusters are located at three distinct loci in the *Azoamicaceae* genomes, with the genes encoding for nitrite reductase and nitric oxide reductase adjacent to each other (Supplementary Data 2). The two most closely related sequences to the *Azoamicaceae narG* gene, encoding for the nitrate reductase, are encoded in genomes of Gammaproteobacteria of the PIVX01 and JACQPQ01 orders. However, it appears that these Gammaproteobacteria likely obtained this gene horizontally from an alphaproteobacterial lineage (Supplementary Fig. 3). The *nirK* gene, encoding for the nitrite reductase is most closely related to uncultured members of the *Ignavibacteraceae* family and was likely horizontally acquired from a member of this clade (Supplementary Fig. 4). The provenance of the nitric oxide reductase and nitrous oxide reductase cannot be conclusively established from the phylogenetic analyzes (Supplementary Figs. 5 and 6), but the *Azoamicaceae norB* gene (encoding for nitric oxide reductase), and to a lesser extent the *nosZ* gene (encoding for nitrous oxide reductase) are related to those found in predatory bacteria of the Bdellovibrionota and Myxococcota phyla, allowing for the possibility that these taxa might have acted as intermediate carriers of these genes. Interestingly, these phylogenetic analyses of the origin of the respiratory genes suggest that the evolutionary model proposed by Graf et al.[7] likely needs to be revised. Unlike originally suggested, the denitrification pathway was most likely absent from the predecessor of the *Azoamicaceae* and only acquired laterally at a later stage. It is possible that the capability to respire nitrogen oxides in addition to oxygen provided the host a competitive advantage for the colonization of micro-oxic and anoxic niches.

Unlike the respiratory genes discussed above, the *tlcA* gene encoding an ATP/ADP transporter, is not monophyletic in the *Azoamicaceae* (Fig. 4b) and seems to have been acquired by the *Azoamicaceae* or the larger UBA6186 order multiple times independently. Both *Azosocius* genomes contain one copy that was acquired by the UBA6186 order from a Chlamydiota organism and vertically inherited by *Azosocius*, but likely lost in *Azoamicus*. The *Azosocius* genomes additionally contain a second copy likely directly acquired from an alphaproteobacterium in the *Rickettsiales* order. The three *Azoamicus* genomes contain a single distinct *tlcA* copy, that also likely originated from a *Rickettsiales* bacterium[7] and is shared with one of the UBA6186 genomes, hinting at the possibility of vertical inheritance and loss in *Azosocius* (Fig. 4b).

## Genetic potential for carbon metabolism and cofactor biosynthesis in *Azoamicaceae*

While the genes encoding the key enzymes of the respiratory pathway are highly conserved across the *Azoamicaceae*, there is less consistency regarding the distribution of genes involved in carbon metabolism across the four genomes (Fig. 2d). Three of the four genomes encode genes for both the oxidation of malate to pyruvate (*maeA*) as well as to oxaloacetate (*mdh*). The *Ca*. A. ciliaticola genome only encodes *mdh*, which was highly transcribed in situ, leading to the suggestion that malate, which is likely provided by the host, acts as the primary electron donor for denitrification[7]. The role of malate as an important metabolite is further supported by the *Ca*. A. aquiferis transcription data analyzed here, showing that *mdh* is the third highest transcribed gene on average, across seven datasets. Conversely, no transcription of *maeA* could be detected in 6 out of 7 datasets (Fig. 3, Supplementary Data 7). Surprisingly, the smallest of the five genomes (*Ca*. A. soli) does not encode either of these enzymes for malate oxidation (Fig. 2d), although it does retain the anion permease proposed to import malate in *Ca*. A. ciliaticola (*yflS*, Supplementary Data 2)[7].

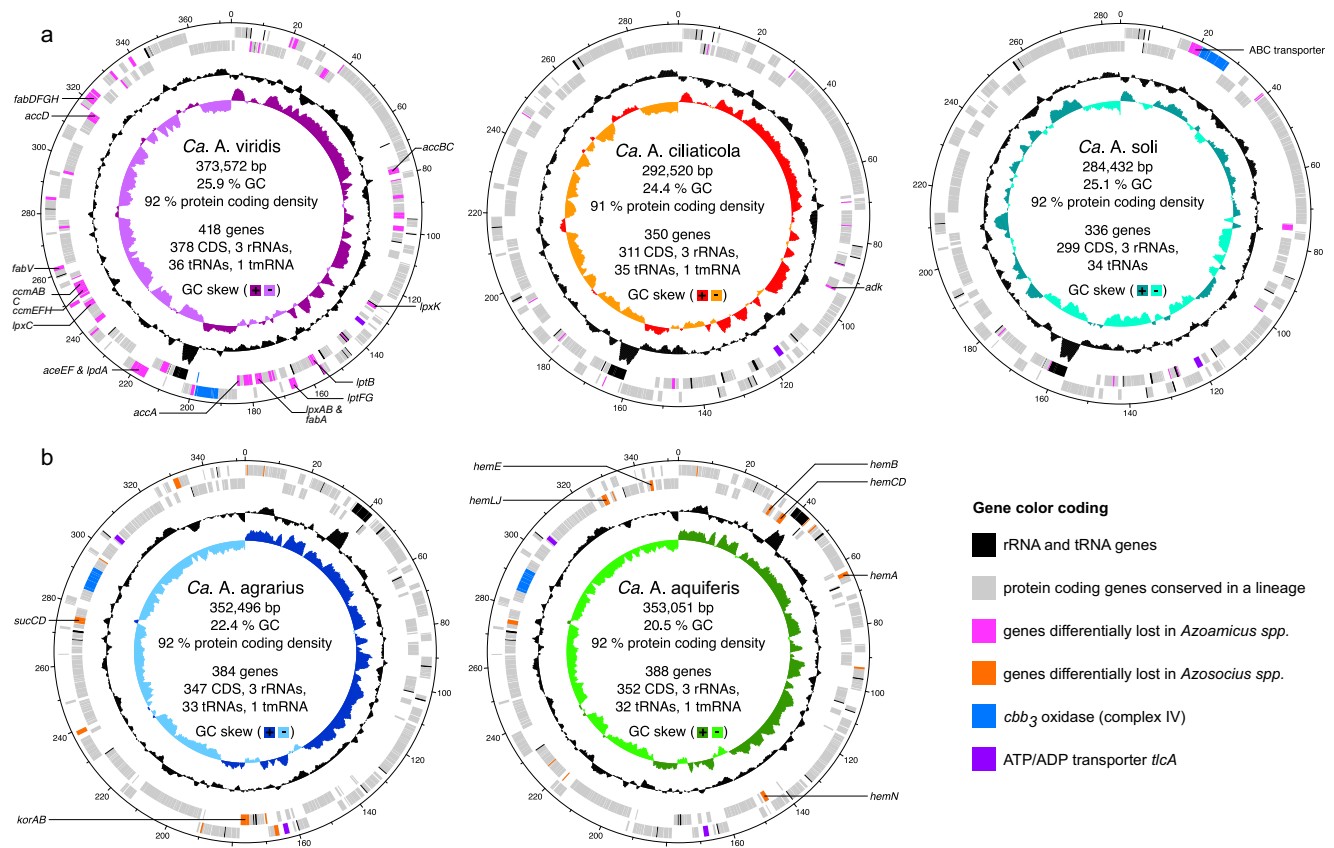

**Fig. 5 | Differential gene loss in *Azoamicaceae* genomes.** Circular genome maps of *Azoamicus* (**a**) and *Azosocius* (**b**) genomes, with rings (inside to outside) indicating GC skew, GC content, reverse strand genes, forward strand genes. rRNA and tRNA genes are indicated in black, protein coding genes encoded in multiple genomes of a genus are indicated in gray. Protein coding genes differentially lost in *Azoamicus* are indicated in pink, and differentially lost in *Azosocius* in orange. The cytochrome *cbb₃* oxidase and associated genes are highlighted in blue, and the ATP/ADP transporter *tlcA* genes in purple.

From the available data, it can not be excluded that *Ca*. A. soli oxidizes malate using host-encoded malate dehydrogenase, but it is also possible that succinate is oxidized by complex II and used as an electron donor instead (Fig. 2d).

Beyond the potential for malate and succinate oxidation, three of the genomes also encode 2-oxoacid:ferredoxin reductase (*korAB*) and succinyl-CoA synthetase (*sucCD*) that can oxidize 2-oxoglutarate to succinyl-CoA, and ultimately to succinate for further oxidation to fumarate using complex II. The *sucCD* genes are lacking in *Ca*. A. soli as well as in *Ca*. A aquiferis, the latter of which also appears to have lost the *korAB* genes (Fig. 2d, Fig. 5), suggesting use of only malate and (possibly) succinate as electron donors, or replacement of korAB and/or sucCD by host proteins.

In addition to genes involved in respiration, other genes conserved in all 5 genomes include those involved in the biosynthesis of metal cofactors. These include the *sufBCDSU* genes for iron-sulfur cluster biosynthesis, as well as the *moaABCDE*, *moeAB*, and *mobA* genes for the assembly of the molybdopterin cofactor required for nitrate reductase (Fig. 2d). In contrast, the *modABC* genes encoding a transporter for molybdenum are lost from the two Azosocius genomes. Interestingly, a near complete gene set for heme biosynthesis (*hemABCDEJLN* and *glnS*) remains in only the *Ca*. A. aquiferis genome, although *hemH* is missing. On the other hand, the cytochrome maturation system encoded by *ccmABCEFH* was found only in *Ca*. A. viridis. Notably, *Ca*. A. viridis also retained the capacity for fatty acid biosynthesis from pyruvate. Malonyl-CoA required for fatty acid biosynthesis can be formed from malate by the activities of malic enzyme (*maeA*), pyruvate dehydrogenase (*aceEF*, *lpdA*), and acetyl-CoA

carboxylase (*accABCD*) (Fig. 5). The latter two complexes are only retained in *Ca*. A. viridis. Fatty acids can then be synthesized from malonyl-CoA by the enzymes of the type II fatty acid biosynthesis pathway (*fabADFGHV*) (Fig. 5)[27]. However, the genes encoding the membrane proteins required to convert fatty acids to phosphatidic acid are missing from *Ca*. A. viridis. Apart from two genes (*cdsA*, *pgsA*) no other genes for synthesis of phosphatidylglycerol or phosphatidylethanolamine were found. Instead, the genome encodes the same *mla* pathway thought to be used for lipid import in the other *Azoamicaceae*[7] (Supplementary Data 2). *Ca*. A. viridis additionally encodes the inner membrane component of the LPS export system (*lptBGF*), as well as several of the steps needed to synthesize lipid-IV_A (*lpxABCDK*). However, the genome is lacking *lpxH*, a gene considered essential for biosynthesis of the Lipid A moiety, as well as key genes of the heptose biosynthesis pathway and the outer membrane component of the export system (*lptADE*), and it seems unlikely that *Ca*. A. viridis can synthesize LPS for a typical outer cell membrane. It is peculiar that the Lipid A biosynthesis pathway has been retained in the strongly reduced genome of *Ca*. A. viridis, suggesting that it may have a yet unidentified role in the symbiosis.

**Plagiopylean ciliates as putative hosts of the *Azoamicaceae***
To identify the potential hosts of the groundwater *Azoamicaceae* we searched for 18S rRNA gene sequences of Plagiopylean ciliates, the known host of *Ca*. A. ciliaticola, in samples from which genomes of the groundwater symbionts were recovered. The source metagenomic data for both *Azosocius* genomes contained two closely related full-length 18S rRNA gene sequences of Plagiopylean ciliates belonging to

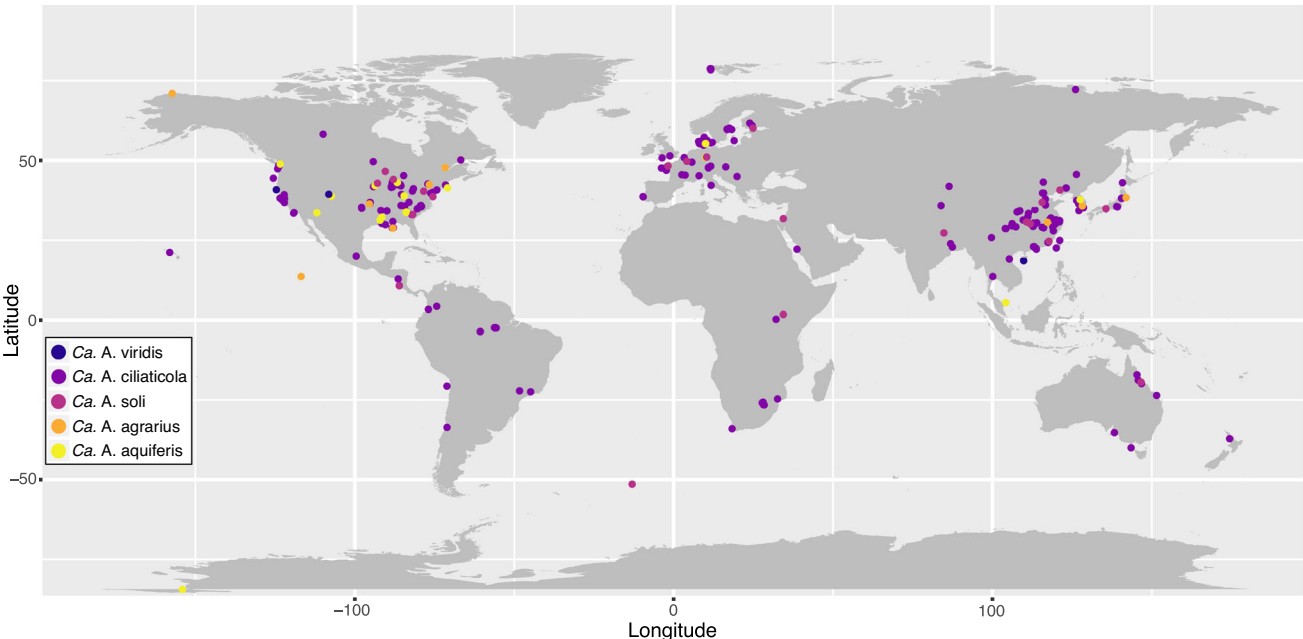

**Fig. 6 | Detection of the *Azoamicaceae* in 16S rRNA gene amplicon sequencing datasets.** Detection of the *Azoamicaceae* 16S rRNA gene sequences in publicly available 16S rRNA gene amplicon sequencing datasets shows a global distribution across all continents. Points represent amplicon datasets, and are colored according to their affiliation with the *Azoamicaceae* genome that have > 95% 16S rRNA gene identity to reads in the dataset. The hotspots in the global north likely reflect sampling bias.

the same order (*Odontostomatida*) as the host of *Ca.* A. ciliaticola (Supplementary Fig. 7a). The combined source datasets for *Ca.* A. viridis and *Ca.* A. soli did not yield a full length Plagiopylean 18S rRNA gene sequence, but targeted assembly of reads mapping to CON-ThreeP 18S rRNA genes resulted in recovery of a fragment representing a partial Plagiopylean ciliate 18S rRNA gene sequence (Supplementary Fig. 7). As the branch containing the putative *Azosocius* hosts in the de novo calculated 18S rRNA gene tree (Supplementary Fig. 7a) has low support, we additionally placed the assembled putative host 18S rRNA gene sequences into a precalculated tree containing only the reference sequences. This approach gave the same topology, suggesting that the added sequences did not affect the branching pattern of the de novo calculated 18S rRNA gene tree (Supplementary Fig. 7b).

## Presence and distribution of *Azoamicaceae* in global amplicon datasets

In addition to comparative genomics, the newly recovered genomes also provide an opportunity to assess the distribution of *Azoamicaceae* respiratory endosymbionts in diverse ecosystems across the world. For this, we searched public 16S rRNA gene amplicon datasets for the presence of the five endosymbiont 16S rRNA gene sequences. We identified 998 unique datasets containing putative *Azoamicaceae* sequences (≥ 3 reads at ≥ 95 % sequence identity). These datasets encompass samples from all continents, suggesting a global distribution of *Azoamicaceae* endosymbionts (Fig. 6). Such global distribution is congruent with the previously estimated early origin of this symbiosis (~106−213 Mya[7]) based on predictions of dispersal of microorganisms[28,29].

It is interesting to note, that despite all four of our new genomes being from groundwater, only 18 samples (2%) containing *Azoamicaceae* reads were annotated as 'groundwater metagenomes' in our global survey. However, the quality of sample type annotation of 16S rRNA gene amplicon datasets is highly variable, and samples of other categories (such as "freshwater metagenome" or "soil metagenome") might encompass more groundwater samples. In any case, the two most common sample types in which putative *Azoamicaceae*

sequences were found were aquatic/freshwater and wastewater/activated sludge (Supplementary Data 8). The previously described *Ca.* A. ciliaticola genome was indeed recovered from anoxic lake water, and our analyzes suggest that symbiont sequences most closely related (≥ 95 % identity) to *Ca.* A. ciliaticola are by far the most common and present in 808 of the 998 datasets (81 %). Relatives of *Ca.* A. agrarius were second most abundant and found in 143 datasets (14 %; Supplementary Data 8). The *Azoamicaceae* sequences were generally rare within the examined datasets, with *Azoamicaceae* reads accounting for ≤ 0.1 % of reads in 90 % of the retrieved datasets. This is consistent with the presence of endosymbionts constrained by host abundance, and may partly explain why the *Azoamicaceae* have only recently been discovered. Reads related to the *Azoamicaceae* accounted for ≥ 1 % of the reads in only 10 out of 998 datasets, which represent wastewater/activated sludge (6), freshwater lake (2), human gut (1), and groundwater (1). In general, our results indicate that wastewater is a widespread and probably important habitat for the symbionts that warrants future attention. Even though the five currently known genomes do not represent the full diversity of the *Azoamicaceae*, they already suggest a ubiquitous presence of the *Azoamicaceae* symbionts in diverse ecosystems around the world.

## Discussion

The four *Azoamicaceae* genomes described in this study greatly expand our understanding of the diversity and distribution of respiratory endosymbiosis. We show that *Azoamicaceae* are globally distributed, and have the genomic potential for both denitrification and aerobic respiration. Of the metabolic capacities encoded in the new endosymbiont genomes it is undoubtedly the capacity to respire oxygen that has the most profound physiological and ecological implications. Retention of the $cbb_3$ terminal oxidase in these extremely reduced genomes, as well as the comparatively high *ccoNOP* transcription in *Ca.* A. aquiferis, strongly implies an active role in endosymbiont and host physiology, by potentially conveying the capacity to respire oxygen to its host. Furthermore, the presence of the $cbb_3$ oxidase can explain the broad distribution of the symbiont in

seasonally or permanently oxic environments, thus considerably expanding the ecological role for this symbiosis. In fact, the capacity for aerobic respiration (or oxygen detoxification) may have facilitated the dispersal of the (presumably anaerobic) host across oxic environments. Endosymbionts are common in anaerobic ciliates, and syntrophic endosymbionts have been proposed to play a role in the transition of ciliates to anaerobiosis[30,31]. Given their potential for aerobic respiration, the *Azoamicaceae* may represent an interesting example of a ciliate endosymbiont facilitating a secondary adaptation, this time to a (micro)aerobic lifestyle.

We propose that Plagiopylean ciliates are the putative hosts of the groundwater *Azoamicaceae*. Indeed, based on a molecular analysis, Ciliophora were abundant in the wells of the Hainich CZE[32], from which *Ca*. A. aquiferis was retrieved. We could reconstruct full length Plagiopylea 18S rRNA gene sequences from the metagenomes containing *Ca*. A. aquiferis as well as *Ca*. A. agrarius. A shorter Plagiopylea 18S rRNA gene fragment from the metagenome containing both *Ca*. A. viridis and *Ca*. A. soli could also be recovered. Interestingly, the phylogeny of 18S rRNA sequences (Supplementary Fig. 7) is congruent with the genome phylogeny of their respective putative symbionts (Fig. 1) and thus could be indicative of long-term vertical inheritance of the symbionts in the *Odontostomatida* order. This is further supported by the extremely reduced nature of the symbiont genomes that is consistent with an older, likely vertically inherited, symbiosis[6]. However, more data on the host identity are needed to confirm an exclusively vertical inheritance of the *Azoamicaceae* symbionts, as some protist hosts are able to replace their symbionts[33,34]. Additionally, while an apparent vertical inheritance of protist endosymbionts can be observed over short evolutionary time[35], examples of long term vertical inheritance are more rare, possibly due to eventual replacement of protist symbionts[6], or simply undersampling[36]. Our comparative genomics of the five *Azoamicaceae* genomes shows a varying degree of genome erosion, as observed in insect HBEs[37,38]. The *Azoamicaceae* seem to converge on a minimal gene set encoding the genes required for ATP generation using a denitrifying and oxygen respiring respiratory chain, with the conserved core genes comprising over 80% of the protein complement of the smallest genomes. The loss of biosynthetic pathways essential for the respiratory function, such as heme biosynthesis and cytochrome c maturation, strongly suggests that host proteins are targeted to the endosymbiont. Host protein targeting has previously been observed in rare cases in insect HBEs[39,40] in the trypanosomatid *Angomonas deanei*[41], and more extensively in the chromatophore of *Paulinella chromatophora*[42] and in the nitroplast (formerly *Candidatus* Atelocyanobacterium thalassa) of *Braarudosphaera bigelowii*[43]. These observations have blurred the boundaries between HBEs and organelles[36,44], and the extant *Azoamicaceae* may represent another snapshot of the transition between the two. While comparative genomic analyzes can provide a great insight into the evolutionary history and metabolic potential of these enigmatic endosymbionts, the establishment of an experimentally tractable laboratory culture of the host is essential to test these exciting hypotheses.

## Method
### Database searching and cMAG curation
The 16S rRNA sequence and nitrate reductase sequence of *Ca*. A. ciliaticola were used to search public genome databases. This led to the identification of the *Candidatus* Azosocius agrarius cMAG consisting of a single circular contig with 100 bp overlap (CP066692), binned from a groundwater sample taken in Modesto, CA, USA (37.661915 N, 121.114554 W) in 2018 (SAMN15459604)[13]. For this study, we removed the overlap and started the contig at the putative origin of replication, resulting in a contig of 352,496 bp. The fasta file with the modified cMAG is provided as Supplementary Data 1.

To identify further samples containing respiratory endosymbionts in groundwater, we used the 16S rRNA gene of *Ca*. A.

agrarius to query publicly available 16S rRNA gene amplicon sequencing data sets using the web search tool IMNGS[45]. This search led to matches with >95% sequence identity in amplicon datasets from a groundwater aquifer located near Hainich national park in Germany, through the long term AquaDiva project (Supplementary Data 8). We screened the assembled metagenomes from the AquaDiva project (51.119338 N, 10.469198 W; Bioproject PRJEB36523)[14] for contigs matching the *Ca*. A. agrarius cMAG using BLASTn[46] (version 2.6.0) with a minimum length cutoff of 3000 bp. We iteratively mapped the trimmed reads from datasets ERR3858113, ERR3858114, ERR3858115 on the resulting 13 contigs using coverM (version 0.7.0) (https://github.com/wwood/CoverM), followed by reassembly of the mapping reads using Spades[47] (version 3.15) with the −isolate flag, resulting in a circular contig of 353051 bp after three iterations, as confirmed by visualization in Bandage[48] (version 0.9.0). This circular contig represents the cMAG of *Candidatus* Azosocius aquiferis and is deposited to genbank under accession number NCBI BioSample SAMN39831648 and NCBI GenBank accession GCA_043390185.1.

For further endosymbiont discovery we constructed a database of proteins encoded by *tlcA* gene, encoding the NTT transporter required for the exchange of ATP and ADP with the host, and screened sequencing runs using a blast score ratio (BSR) approach as previously described[49,50]. We then queried the SRA for groundwater datasets, and selected large and medium scale BioProjects to screen. We screened a combined 256 sequencing runs from BioProjects PRJNA640378, PRJEB36505, PRJEB28738, PRJNA530103, PRJNA268031, PRJNA292723, PRJEB32173, PRJEB14718, PRJNA513876, and PRJNA512237 encompassing groundwater samples from Australia, Saudi Arabia, Germany, and USA (Colorado, Tennessee, California and Ohio). While we found evidence for the presence of putative endosymbionts (at low abundance) in several datasets, we only managed to reconstruct additional cMAGs, which was the quality threshold we set for this study, from sequencing runs from bioproject PRJNA512237[16]. cMAG reconstruction for *Ca*. A. viridis and *Ca*. A. soli was performed by first co-assembling trimmed reads from datasets SRR8863434, SRR8863435, and SRR8863439 using MEGAHIT[51] (version 1.2.9). The resulting contigs were searched using BLASTn[46] (version 2.6.0) using the three other cMAGs as query, keeping hits with a minimum length cutoff of 3000 bp. This approach yielded 3 contigs, a circular contig representing the cMAG of *Candidatus* Azoamicus soli, and two contigs forming the MAG of *Candidatus* Azoamicus viridis. The *Ca*. A. viridis MAG was then circularized as described above for *Candidatus* A. aquiferis. The cMAG of *Candidatus* Azoamicus soli is available via NCBI BioSample SAMN39831885 and NCBI GenBank accession GCA_043390205.1 and the cMAG of *Candidatus* Azoamicus viridis is available via NCBI BioSample SAMN39831884 and NCBI GenBank accession GCA_043390165.1.

### Genome annotation and comparative genomics
cMAGs were analyzed using anvi'o[52] (version 7.1), with gene calling by prodigal[53] (version 2.6.3). Annotation of protein coding genes was transferred from *Ca*. A. ciliaticola where appropriate based on bidirectional best hit DIAMOND (version 2.0.14.152)[54], and the genes were annotated using KEGG[55], COG[56] and PFAM[57] annotation as integrated in anvi'o, followed by manual curation of the annotation. The final annotation is provided as Supplementary Data 2. The UpSet plot[58] of the cMAG gene content was generated from a presence absence matrix of the genes content using the UpSetR R package (version 1.4.0)[59].

The average nucleotide identity between the cMAGs was calculated using fastANI[17] (version 1.33) with a fragment length of 1000 basepairs. Average amino acid identity was calculated using ezAAI[60] (version 1.2.2) with default settings. 16S rRNA genes were extracted from the cMAGs using anvi'o (development version)[52], and pairwise identity was calculated using BLASTn (version 2.6.0)[46].

Metabolic prediction of the cMAGs was done using the anvi-estimate-metabolism program (https://anvio.org/help/main/programs/

anvi-estimate-metabolism/) as integrated in anvi'o[52] (development version) using KEGG modules annotation.

## Pseudogene and single nucleotide variant detection

Pseudogenes in the Azoamicaceae genomes were detected using pseudofinder (version 1.1.0)[61]. The Azoamicaceae genomes were reannotated using prokka (version 1.14.6)[62] due to required genbank file format input. Predicted proteins and intergenic regions were used as BLASTp and BLASTx (version 2.12.0 +)[46] queries against the NCBI-nr protein database (downloaded 11 June 2024). This resulted in 13–16 predicted pseudogenes per genome. The gene coordinates of predicted pseudogenes were compared with the anvi'o annotation, and pseudogenes consistent across both annotation methods (3–8 per genome) were added to Supplementary Data 5. Single nucleotide variation in the five Azoamicaceae genomes was assessed using inStrain (version 1.9.0)[63] after read mapping with coverM (version 0.7.0) (https://github.com/wwood/CoverM) requiring mapped reads to have > 95 % identity over > 80 % of the read length. Results for analyzed dataset and reference pairs are summarized in Supplementary Data 6.

## Metatranscriptome analysis

The 18 metatranscriptome sequencing read datasets in BioProject PRJEB28738 were retrieved from NCBI and trimmed using the cutadapt (version 3.7)[64] wrapper trim-galore (version 0.6.10) (https://github.com/FelixKrueger/TrimGalore) with default settings and automatic adapter detection. The trimmed reads were mapped against all gene sequences in the Ca. A. aquiferis genome using coverM (version 0.7.0) (https://github.com/wwood/CoverM), with minimum identity 95 % and minimum length fraction 80 %, exporting the read counts per gene sequence. Based on total reads mapped, seven datasets were selected for analysis (Supplementary Data 7). Read counts matching protein coding genes were used for the calculation of transcript per million (TPM) values[65] and genes were ranked according to average TPM across the seven datasets. For visualization of the transcriptome in a heatmap (Fig. 3), transcript abundance was recalculated as "transcript per thousand", following the same procedure as for TPM but using a $10^3$ scaling factor instead of $10^6$ to account for the number of reads mapped to Ca. A. aquiferis. Datasets included in the heatmap are ERR2809165 (H52_1), ERR2809163 (H52_2), ERR2809153 (H52_3), ERR2809156 (H41_1), ERR2809155 (H41_2), ERR2809154 (H41_3), and ERR2809148 (H41_4). Samples H5-2_1, H5-2_2, and H5-2_3 represent replicated metatranscriptome data from the same well (H5-2) from sampling campaign PNK69. Sample H4-1_4 represents a metatranscriptomic data set from three pooled RNA preparations from well H4-1 from from sampling campaign PNK66.

## Global detection of Azoamicaceae

The 16S rRNA genes from the four cMAGs described here, and the previously published Candidatus Azoamicus ciliaticola, were used to screen 16S rRNA gene amplicon datasets for matches using the IMNGS web search tool[45]. 998 Datasets with more than 3 reads matching any of the five cMAGs at ≥ 95 % identity were retained for further analysis. The dataset IDs were used to retrieve sample metadata from NCBI, and sample coordinates were manually standardized. Samples with coordinates were plotted on world map (Fig. 6) in R using packages maps (https://CRAN.R-project.org/package=maps) and mapdata (https://CRAN.R-project.org/package=mapdata). For 210 of the 998 datasets no coordinates were provided and these are therefore not included in Fig. 6. To assess environmental distribution of the Azoamicaceae, the 998 datasets were grouped by their dataset description (defined upon dataset submission to INSDC databases).

## Putative host detection

To gain insight into potential host organisms for the organisms represented by the cMAGs, we attempted to reconstruct full length 18S

rRNA gene sequences from their source datasets using Phyloflash (version 3.4.2)[66]. One 18S rRNA gene sequence was from each of ERR3858113, ERR3858114, ERR3858115 (source data for Ca. A. aquiferis). These three sequences were identical, and belonged to a ciliate of the Plagiopylea class. Four 18S rRNA gene sequences were retrieved from SRR12113421 (source data for Ca. A. agrarius). One of the four sequences obtained from SRR12113421 belonged to a ciliate of the Plagiopylea class, closely related to the sequences obtained from ERR3858113, ERR3858114, ERR3858115. No full length 18S rRNA gene sequences were retrieved from datasets SRR8863434, SRR8863435, and SRR8863439 (source data for Ca. A. viridis and Ca. A. soli), but reads matching CONthreeP 18S sequences were detectable using PhyloFlash. Fasta sequences of the reads matching CONthreeP were extracted from the PhyloFlash output of SRR8863434, SRR8863435, and SRR8863439, combined, and assembled using idba-ud (version 1.1.3)[67], resulting in one contig representing a partial Plagiopylean 18S rRNA gene sequence.

## 18S rRNA gene phylogeny

The four recovered (partial) putative host 18S rRNA gene sequences were used as input for the web based Silva alignment, classification, and tree service (Silva-ACT https://www.arb-silva.de/aligner/). The maximum number of neighbor sequences was set to 100 (maximum) and the identity was set to 75%. Alignment was done with the SINA aligner (version 1.2.12)[68] with the variability profile for Eukaryota. A de novo tree with neighbor sequences was computed using Fasttree[69] (no version provided in Silva-ACT web interface) with the GTR model and Gamma likelihood rate model with the stringent positional variability filter for Eukarya. The resulting tree was visualized in iToL[70]. To support the obtained topology, a second tree was generated using the Silva-ACT web interface, by placing the four putative host sequences in the tree of neighbors using RaxML[71] (no version provided in Silva-ACT web interface) and the resulting tree was also visualized in iToL.

## Phylogenetic analyzes of individual and concatenated amino acid sequences

Phylogenetic analysis of protein coding genes (nuoG, sdhA, qcrB, atpA, ccoN, narG, nirK, norB, nosZ, tlcA) was done using protein sequences obtained from genomes in the genome taxonomy database (GTDB, version 207)[72] and the genomic catalog of Earth's microbiomes[73]. Sequences were aligned using MUSCLE[74] (version 3.8.31), and phylogenetic trees were calculated using IQ-tree[75] (version 2.2.0) with automatic model selection using ModelFinder[76] (integrated in IQ-tree, no separate version provided) and ultrafast bootstrapping (1000 replicates) with UFBoot2[77] (integrated in IQ-tree, no separate version provided). Models selected by ModelFinder were: NuoG - Q.pfam+R10; SdhA - LG+I+I+R10; QcrB - LG+F+I+I+R10; AtpA - Q.yeast+I+I+ R10; CcoN - Q.pfam+I+I+R10; NarG - Q.yeast+I+I+R7; NirK - Q.pfam +I+I+R9; NorB - Q.yeast+F+I+I+R6; NosZ - LG+F+I+I+R7; TlcA - Q.yeast+F+R10).

Concatenated marker gene phylogenies were calculated based on protein sequences identified using hmmer[78] (version 3.3.2) with the Bacteria_71 HMM set included in Anvi'o[52] (version 7.1). The anvi-get-sequences-for-hmm-hits program was used to retrieve sequences for HMM hits, align them using MUSCLE[74] (version 3.8.31), concatenate alignments, and write a partition file. The concatenated alignments and corresponding partition files were used to calculate phylogenies using IQ-tree[75] (version 2.2.0) with automatic model selection using ModelFinder[76] (integrated in IQ-tree, no separate version provided) and ultrafast bootstrapping (1000 replicates) with UFBoot2[77] (integrated in IQ-tree, no separate version provided). All data for the phylogenetic analyzes are available on figshare at https://figshare.com/projects/Groundwater_Azoamicaceae_phylogenies/205756.

## Etymology

### Description of 'Candidatus Azoamicus viridis' sp. nov

'Candidatus Azoamicus viridis' (L. masc. adj. viridis, green pertaining to Greene county, where the sample containing the species was taken). A bacterial species identified by metagenomic analyzes. This species includes all bacteria with genomes that show ≥ 95% average nucleotide identity to the type genome for the species to which is available via NCBI BioSample SAMN39831884 and NCBI GenBank accession GCA_043390165.1.

### Description of 'Candidatus Azoamicus soli' sp. nov

'Candidatus Azoamicus soli' (L. masc. n. soli, of the earth). A bacterial species identified by metagenomic analyzes. This species includes all bacteria with genomes that show ≥ 95% average nucleotide identity to the type genome for the species to which is available via NCBI BioSample SAMN39831885 and NCBI GenBank accession GCA_043390205.1.

### Description of 'Candidatus Azosocius' gen. nov

'Candidatus Azosocius' (N.L. masc. n. Azosocius, combines the prefix azo- (N. L., pertaining to nitrogen) with socius (Latin, masc. n., associate); thus giving azosocius ('associate that pertains to nitrogen'). A bacterial genus identified by metagenomic analyzes and delineated according to Relative Evolutionary Distance by the Genome Taxonomy Database (GTDB). The type species of the genus is 'Candidatus Azosocius agrarius'.

### Description of 'Candidatus Azosocius agrarius' sp. nov

'Candidatus Azosocius agrarius' (L. masc. adj. agrarius, of the soil). A bacterial species identified by metagenomic analyzes. This species includes all bacteria with genomes that show ≥ 95% average nucleotide identity to the type genome for the species to which is available via NCBI BioSample SAMN15435421 and NCBI GenBank accession GCA_016432505.1.

### Description of 'Candidatus Azosocius aquiferis' sp. nov

'Candidatus Azosocius aquiferis' (N.L. gen. masc. n. aquiferis, of an aquifer). A bacterial species identified by metagenomic analyzes. This species includes all bacteria with genomes that show ≥ 95% average nucleotide identity to the type genome for the species to which is available via NCBI BioSample SAMN39831648 and NCBI GenBank accession GCA_043390185.1.

### Description of 'Candidatus Azoamicaceae' fam. nov

'Candidatus Azoamicaceae' (A.zo.a.mi.ca.ce.ae. N.L. masc. n. Azoamicus. type genus of the family; N.L. suff. –aceae to denote a family; N.L. fem. pl. n. Azoamicaceae, the family of the genus Azoamicus). The description of the family 'Candidatus Azoamicaceae' is the same as that of the genus 'Candidatus Azoamicus'. The type genus is 'Candidatus Azoamicus'.

### Description of 'Candidatus Azoamicales' ord. nov

'Candidatus Azoamicales' (A.zo.a.mi.ca.les. N.L. masc. n. Azoamicus. type genus of the order; N.L. suff. –ales to denote an order; N.L. fem. pl. n. Azoamicales, the order of the genus Azoamicus). A bacterial order identified by metagenomic analyzes and delineated according to Relative Evolutionary Distance by the Genome Taxonomy Database (GTDB). Phylogenetic analyzes in this work have shown that the previously published 'Candidatus Azoamicus ciliaticola' is assigned to an uncharacterized order with provisional designation UBA6186. Consequently, we propose to rename the UBA6186 order to 'Candidatus Azoamicales'. the type genus of the order is 'Candidatus Azoamicus'. The order is assigned to the class Gammaproteobacteria.

### Reporting summary

Further information on research design is available in the Nature Portfolio Reporting Summary linked to this article.

## Data availability

The sequencing data used to construct the symbiont genomes described in this study is publicly available in Genbank under Bio-Project numbers PRJEB36523 and PRJNA512237. The successfully reconstructed genomes are available in Genbank under Bioproject number PRJNA1073475 and accession numbers GCA_043390165.1, GCA_043390205.1, and GCA_043390185.1. The previously reconstructed genome that we identified as Candidatus Azosocius agrarius is available in Genbank under accession number GCA_016432505.1. Metatranscriptome sequencing data used was publicly available in Genbank under BioProject accession number PRJEB28738. 16S rRNA gene amplicon data was accessed through the IMNGS web server (https://www.imngs.org/). Data files for phylogenetic trees discussed in the manuscript, as well as the anvi'o generated genome annotation files used to generate Supplementary Data 2 are available on figshare at https://doi.org/10.6084/m9.figshare.c.7510170.v1. Source data for all other figures can be found in Supplementary Data files 1-8.

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

## Acknowledgements

We thank Falko Gutmann and Robert Lehmann for assistance with sampling groundwaters at the Hainich CZE as part of the Collaborative Research Center AquaDiva. This study used publicly available data from Bioprojects PRJNA640378, PRJEB36523, and PRJNA512237 and we thank the authors of those studies for making their data available. This study was financially supported by the Max Planck Gesellschaft. This research was further supported by the Deutsche Forschungsgemeinschaft (DFG) under Germany's Excellence Strategy—EXC 2051, Project-ID 390713860 (to K.K.) and the Collaborative Research Center AquaDiva (CRC 1076 AquaDiva - Project-ID 218627073, to K.K) of the Friedrich Schiller-University Jena.

## Author contributions

D.R.S. and J.M. designed research. D.R.S. and L.M.Z. organized and performed sampling at the Hainich critical zone exploratory (CZE) through the Collaborative Research Center (CRC) AquaDiva and analyzed field sampling data. J.S.G performed initial 16S rRNA gene surveys and identified the *Ca*. Azosocius agrarius genome. K.K. and W.A.O. contributed data from previous sampling campaigns by CRC AquaDiva at the Hainich CZE. D.R.S. processed, synthesized, and analyzed sequence data. D.R.S. and J.M. interpreted genomic data. D.R.S. and J.M. wrote the manuscript with contributions from all authors.

## Funding

## Competing interests

The authors declare no competing interests.
