## [Transparent Peer Review file · Nature Communications]

Genetic potential for aerobic respiration and denitrification in globally distributed respiratory endosymbionts

Corresponding Author: Dr Daan Speth

Version 0:

Reviewer comments:

Reviewer #1

(Remarks to the Author)

Speth et al., searched for and investigated genomes from close relatives of *Ca. Azoamicus ciliaticola*, an intracellular symbiont of a plagiopylean ciliate that allegedly provides ATP to its host by nitrate respiration. They identify several close relatives which reveal that *Ca. Azoamicus ciliaticola* was ancestrally both an aerobic and anaerobic respirer. The authors furthermore examine metagenomes from across the world and find that *Ca. Azoamicus ciliaticola* and close relatives have a broad distribution.

The study is sound and the methods adequate. The manuscript is clear and well written. I do not have any major objections. The question asked is straightforward and the conclusions about the ancestry of *Ca. Azoamicus ciliaticola* are as well supported. The conceptual advance of the discovery is, however, rather limited. The authors had previously discussed the possibility that *Ca. Azoamicus ciliaticola* was ancestrally aerobic, and the worldwide distribution of *Ca. Azoamicus ciliaticola* and relatives is not surprising as similar patterns have been found for many other rare microbes, and is to be expected due to the high dispersal rate of unicellular organisms ("everything is everywhere, the environment selects"). This latter point could be further discussed by the authors.

I think the constant and confident statements of an ATP supplying capacity by the intracellular symbionts have to be toned down. Although I agree the authors, and believe, that the symbionts may export ATP, this has not yet been convincingly demonstrated by the authors in this or their previous study.

Reviewer #2

(Remarks to the Author)

Review for NCOMMS-24-12219

Genetic potential for aerobic respiration and denitrification in globally distributed respiratory endosymbionts

Speth and colleagues report novel circularized metagenomic assembled genomes related to the respiratory endosymbiont *Candidatus Azoamicus ciliaticola*. The results point to additional roles for this style of symbiont in different environmental contexts. Even though the symbionts have reductively have arrived at a similar set of genes they still have clear differences, some of which might be compensated by the host. The identification of cytochrome *cbb3* oxidase provides the presumptive host cell the ability to respire oxygen in addition to nitrate. The origin of important genes (nitrogen and oxygen respiration, nucleotide transport) appears to be via lateral gene transfer, in some cases showing repeated transfers of the same genes from different microbial lineages. This is exciting and shows that reduced symbionts are not excluded from gene transfer events. The authors are able to recover gene expression data for these genomes that supports the importance of both nitrogen and oxygen respiratory genes. The host organism of the respiratory symbiont is unknown but the data is consistent with it being a ciliate adapted to microaerophilic conditions.

The finding and definition of host-beneficial endosymbionts and their relationship to their hosts is a highly topical and exciting field of research. The results are presented clearly and concisely and are of a very high standard. The methodology is sound and represents state-of-the art in the field. The work supports the claims of the authors. The work is a great addition to its field and I support its publication in Nature Communications with the appropriate revisions suggested below. I have two

major points of correction to do with the phylogenetic analyses and data availability and some minor points of correction.

Major point:

Line 500: The authors mention that IQtree and ModelFinder was used to select the phylogenetic model to for phylogenetic reconstruction. However, I was not able to find any record of what different models that were deemed optimal and actually applied for each reconstruction. This needs to be provided for each phylogeny either in the respective legend or in the Material and methods section. Also, the alignments for the phylogenies have not been made available for review and needs to be made available via a repository (Figshare, Zenodo or similar) for evaluation and data transparency reasons.

Line 412, 434, 435, 524, 530, 549 : <Accession number pending>; accession numbers needs to be added for all the *Azoamicus* spp. and *Azosocius* spp. genomes.

Minor points

I don't see any explicit mention about pseudogenes in any of the genomes. If none are present then I would like to see that mentioned. Otherwise I would like to see a scan of the genomes to identify potential pseudogenes in intergenic regions.

How much single nucleotide variation was present in the cMAGs? Is there evidence of microvariation in the different *Azoamicus* spp. and *Azosocius* spp. genomes.

Line 351: *cbb3* in italics

Line 372: I would recommend also referencing the nitroplast in *B. bigellowii* here. T. H. Coale et al., Science 384, 217 (2024).

Line 383: Two punctations, is something missing here?

Figure 4, Extended data figure 3, 4, 5, 6: The wedges designating collapsed part of the phylogenetic tree are not adequately annotated, please add descriptions on what was collapsed in the figure and in the legend.

Reviewer #3

(Remarks to the Author)

This manuscript describes four new complete MAGs of putative denitrifying symbionts of ciliate protists. These MAGs were retrieved from publicly available sequencing projects, but presence of their putative hosts has not been demonstrated. I think the study is interesting to the field of protistology and microbial interactions in general. The work is well done and I do not have major concerns about the methodology. Some additional analyses and considerations of the host are needed to really push this study one step further. However, the lack of a tractable system will make further advances challenging.

I have a few suggestions to help improve the manuscript:

INTRO

- Line 42 implies the host has lost its mitochondria. This is incorrect. The mitochondria has become 'specialized' for life without oxygen. These are often referred to as 'metabolically reduced' or metabolically retailed' mitochondria often known as hydrogenosomes. Some in the field prefer the term 'mitochondrion-related organelles' as the term 'hydrogenosome' requires that hydrogen production has been experimentally generated.

INFORMATION ABOUT THE HOST IS LACKING

- The results section lacks any mention of searches for the host 18S but it is implied in the methods and discussion. Consider adding a sentence or two about this in the results.

- Similarly, were attempts to identify ciliate transcripts in the metatranscriptome? Perhaps the 18S is hiding in there or even some obvious eukaryotic transcripts (e.g., actin, tubulin). Ciliates can have a different genetic code, so this is often used to identify ciliate transcripts.

- Is there any speculation about what the host might provide to the symbiont? H₂? Are there H₂-uptake hydrogenases still encoded by the symbiont?

Considerations about the host:

- There are dozens of papers out now speculating on the nature of protist:prokaryote symbioses, and indeed in the ciliates it's arguably the most well characterised or speculated upon. So the present manuscript leaves me wanting to know more about what the host is doing (to be fair, this information was sparse in the original paper describing this particular system likely because its EXTREMELY difficult to retrieve information about the host). But one 'easy'-ish way to test for this would be to look for eukaryotic transcripts in the metatranscriptomic assemblies The authors discussed the presence of the 18S, but I wonder if there are other transcripts expressed from the host that could link in with the genes being expressed in the

bacterium. To do this, you would have to assemble the transcriptomes and do 'tblastn' with candidate genes important for the symbiosis. These could be the hydrogenase genes that are quite unique in the ciliates compared to other eukaryotes (see Boxma et al. 2007 BMC eco evol) or other important genes that have been described in recent studies describing the hydrogenosomal/MRO proteomes of anaerobic ciliates and their relationships to their symbionts (e.g., Lewis et al. MBE 2020; Lind et al. ISME J 2018, Rotterova et al. Current Biology 2020). The ciliates sometime have an alternative genetic code that you can use for deciphering bacterial vs eukaryotic transcripts (run it with codon table =6) . As an aside, I tried to run these BLASTs myself, but the assembly was taking too long...

- It seems like the authors could only find 2 different 18S rRNA sequences belonging to ciliates in the different transcriptome projects which makes constructing phylogenetic trees challenging. Were there any attempts made to resolve these 18S sequences in a larger phylogeny of ciliates? It would be interesting to see if the ciliates and the symbionts have co-evolved like has been previously reported in other ciliates (Lind et al. 2018 ISME J) or if there has been 'symbiont swapping' that has been reported as well (Boscaro et al. 2022 Current Biology) and reviewed broadly in (Husnik et al. 2021 Current biology).

- In general, a lot of protist literature is overlooked and it could be worth looking at some of the hypothesis pieces that have come out about ciliate symbioses (to complement the insect systems that are currently referenced). For example, Rotterova et al J Euk Micro 2022.

GENERAL

- Line 45 is redundant to line 42.

- Line 99 - why is it striking that the genomes are the same size?

- are the cMAGs closed genomes? or linear?

Version 1:

Reviewer comments:

Reviewer #2

(Remarks to the Author)

All of my points have been addressed, thank you to the authors for that, and congratulations on your great contribution.

Reviewer #3

(Remarks to the Author)

The authors have satisfied my concerns and I think addressed many of the concerns of the other reviewers.

Two comments:

Its interesting to hear that the potential interaction is NOT based on hydrogen exchange (i missed this in my first read through of Graf et al. years ago); I would guess that the MRO has nothing to do with the symbiosis then unless malate is exported from the organelle (possible)? Just out of curiosity, did you ever see the MRO near the symbiont (as has been reported in other ciliate systems)?

Just a note for future projects, there are not a lot of protist sequences available on genbank, so its not so surprising that attempts to find ciliate homologues came up short. If you haven't already checked, perhaps the EukProt v3 database could be of use - they have lots of predicted proteomes (blast server) and some transcriptomes (figshare) from ciliates including two Plagiopyla.

Response to reviewer comments

Reviewer #1

Septe et al., searched for and investigated genomes from close relatives of *Ca. Azoamicus ciliaticola*, an intracellular symbiont of a plagiopylean ciliate that allegedly provides ATP to its host by nitrate respiration. They identify several close relatives which reveal that *Ca. Azoamicus ciliaticola* was ancestrally both an aerobic and anaerobic respirer. The authors furthermore examine metagenomes from across the world and find that *Ca. Azoamicus ciliaticola* and close relatives have a broad distribution.

The study is sound and the methods adequate. The manuscript is clear and well written. I do not have any major objections. The question asked is straightforward and the conclusions about the ancestry of *Ca. Azoamicus ciliaticola* are as well supported.

The conceptual advance of the discovery is, however, rather limited. The authors had previously discussed the possibility that *Ca. Azoamicus ciliaticola* was ancestrally aerobic, and the worldwide distribution of *Ca. Azoamicus ciliaticola* and relatives is not surprising as similar patterns have been found for many other rare microbes, and is to be expected due to the high dispersal rate of unicellular organisms (“everything is everywhere, the environment selects”). This latter point could be further discussed by the authors.

We thank the reviewer for their evaluation and we have now revised the manuscript to better highlight the novelty and originality of the data presented in our study. While the reviewer is right that the facultative aerobic lifestyle was discussed previously, at that point in time, this was only a speculation. Moreover, our finding that the terminal oxidase as well as the denitrification genes were in fact obtained horizontally was unexpected, and suggests that – unlike the originally proposed evolutionary model – *Ca. A. ciliaticola* likely derived from a non-denitrifying ancestor. We have now clarified this further in the results section at lines 276-282.

We respectfully disagree that the global distribution of this symbiosis is not surprising, but irrespective of that, this first report of the distribution of the Azoamicaceae (and their hosts) is essential for evaluating the ecological role of this symbiosis. As suggested, we have now added a discussion about the dispersion of these symbionts and a speculation about their contribution to the ecological success of their host to the results section at line 411-413.

I think the constant and confident statements of an ATP supplying capacity by the intracellular symbionts have to be toned down. Although I agree the authors, and believe, that the symbionts may export ATP, this has not yet been convincingly demonstrated by the authors in this or their previous study.

We agree with the reviewer that the ATP supplying function of *Ca. Azoamicus ciliaticola* hasn't been directly demonstrated yet. Therefore, we have revised, removed, or toned down the respective statements at lines 24, 52, 78, 91, 167, and 229.

Reviewer #2

Speth and colleagues report novel circularized metagenomic assembled genomes related to the respiratory endosymbiont *Candidatus Azoamicus ciliaticola*. The results point to additional roles for this style of symbiont in different environmental contexts. Even though the symbionts have reductively have arrived at a similar set of genes they still have clear differences, some of which might be compensated by the host. The identification of cytochrome *cbb3* oxidase provides the presumptive host cell the ability to respire oxygen in addition to nitrate. The origin of important genes (nitrogen and oxygen respiration, nucleotide transport) appears to be via lateral gene transfer, in some cases showing repeated transfers of the same genes from different microbial lineages. This is exciting and shows that reduced symbionts are not excluded from gene transfer events. The authors are able to recover gene expression data for these genomes that supports the importance of both nitrogen and oxygen respiratory genes. The host organism of the respiratory symbiont is unknown but the data is consistent with it being a ciliate adapted to microaerophilic conditions.

The finding and definition of host-beneficial endosymbionts and their relationship to their hosts is a highly topical and exciting field of research. The results are presented clearly and concisely and are of a very high standard. The methodology is sound and represents state-of-the art in the field. The work supports the claims of the authors. The work is a great addition to its field and I support its publication in Nature Communications with the appropriate revisions suggested below. I have two major points of correction to do with the phylogenetic analyses and data availability and some minor points of correction.

We thank the reviewer for their kind comments.

Major point:

Line 500: The authors mention that IQtree and ModelFinder was used to select the phylogenetic model to for phylogenetic reconstruction. However, I was not able to find any record of what different models that were deemed optimal and actually applied for each reconstruction. This needs to be provided for each phylogeny either in the respective legend or in the Material and methods section. Also, the alignments for the phylogenies have not been made available for review and needs to be made available via a repository (Figshare, Zenodo or similar) for evaluation and data transparency reasons.

We apologize for this omission in our method reporting. We have now added the applied models to the methods section at lines 691-694.

We have also created a figshare repository where we have deposited the (aligned) protein data sets, as well as the files for trees included in the manuscript. The repository can be accessed at [https://figshare.com/projects/Groundwater_Azoamicaceae_phylogenies/205756](https://figshare.com/projects/Groundwater_Azoamicaceae_phylogenies/205756) and this link has been included in the methods section at line 702-703.

Line 412, 434, 435, 524, 530, 549 : ; accession numbers needs to be added for all the Azoamicus spp. and Azosocius spp. Genomes.

We apologize for this oversight; we have now added the accession numbers for all Azoamicus/Azosocius genomes to the revised manuscript, in both the methods and etymology section (at lines 558, 581, 582, 712, 718, and 739). We want to clarify that the described genomes were submitted to the NCBI prior to original submission of the manuscript but the genome accession numbers only became available during the review process.

Minor points

I don't see any explicit mention about pseudogenes in any of the genomes. If none are present then I would like to see that mentioned. Otherwise I would like to see a scan of the genomes to identify potential pseudogenes in intergenic regions.

Thank you for bringing up this interesting point. We have now run the pseudogene detection tool pseudofinder, which looks for pseudogenes in both the fragmented genes still recognized by prodigal, as well as in the intergenic regions. The exact details of the method are now included in the Material and Methods section at lines 603-613.

There are 3-8 predicted pseudogenes in the Azoamicaceae genomes, which we list in a new Supplementary Table S5. We now also discuss pseudogenes in the results section at line 139-141.

How much single nucleotide variation was present in the cMAGs? Is there evidence of microvariation in the different Azoamicus spp. and Azosocius spp. Genomes.

There is a small amount of microvariation present in the Azoamicaceae genomes, but the number of SNVs is very limited (0-65 SNV detected in the individual genomes using InStrain). However, we would like to point out that our requirement for circular genomes from our genome recovery approach might have selected for datasets that have low strain level diversity, and therefore assemble more readily into single contigs.

Additionally, the low number of SNVs may be caused by the low coverage of the Azoamicaceae in the source datasets. For example, the higher SNV numbers found in some of the *Ca. Azoamicus ciliaticola* samples that were more deeply sequenced indicate that there can be strain level variation in the Azoamicaceae populations, but a clear correlation between sequencing depth and number of SNVs is lacking.

We have added the results of our new InStrain analyses to the results (line 143-152) and methods sections (line 613-617) and added a summary of the data as Supplementary Table S6

Line 351: *cbb3* in italics

This has been changed

Line 372: I would recommend also referencing the nitroplast in *B. bigelowii* here. T. H. Coale et al., Science 384, 217 (2024).

This reference has been added.

Line 383: Two punctations, is something missing here?

Nothing was missing, the extra period was added by accident. This has been removed.

Figure 4, Extended data figure 3, 4, 5, 6: The wedges designating collapsed part of the phylogenetic tree are not adequately annotated, please add descriptions on what was collapsed in the figure and in the legend.

Thank you for pointing out this omission, we have now added names and descriptions of the collapsed part of the phylogenies to the figures and the figure legends.

Reviewer #3

This manuscript describes four new complete MAGs of putative denitrifying symbionts of ciliate protists. These MAGs were retrieved from publicly available sequencing projects, but presence of their putative hosts has not been demonstrated. I think the study is interesting to the field of protistology and microbial interactions in general. The work is well done and I do not have major concerns about the methodology. Some additional analyses and considerations of the host are needed to really push this study one step further. However, the lack of a tractable system will make further advances challenging.

We thank the reviewer for their kind review. We agree that we did not address the host identity of the new symbiont lineages in much detail.

Indeed, without a culture our possibilities are somewhat limited, nonetheless, we have now added new molecular (18S rRNA gene tree) data that allow us to discuss the identity of the putative host of the groundwater lineages. Additionally, we have substantially extended the discussion of the available protist literature. These additions are discussed in more detail below.

I have a few suggestions to help improve the manuscript:

INTRO

- Line 42 implies the host has lost its mitochondria. This is incorrect. The mitochondria has become 'specialized' for life without oxygen. These are often referred to as 'metabolically reduced' or 'metabolically retailed' mitochondria often known as hydrogenosomes. Some in the field prefer the term 'mitochondrion-related organelles' as the term 'hydrogenosome' requires that hydrogen production has been experimentally generated.

Thank you for pointing this out.

We have now changed this sentence:

“The Plagiopylean ciliate host of *Ca. A. ciliaticola* appears to have lost its mitochondria, and is thus incapable of (aerobic) respiration, although it might still be able to generate ATP through fermentation using hydrogenosomes, or substrate level phosphorylation in the cytoplasm”

To (revised manuscript line 53-57):

“The Plagiopylean ciliate host of *Ca. A. ciliaticola* was proposed to harbor only metabolically reduced organelles known as mitochondrion-related organelles (MROs) (Müller Miklós et al. 2012; Stairs et al. 2015), possibly in the form of hydrogenosomes. Thus the host is likely incapable of (aerobic) respiration, although it might still be able to generate ATP in the MROs, or through substrate level phosphorylation in the cytoplasm. (Embley et al. 1997; Graf et al. 2021).”

INFORMATION ABOUT THE HOST IS LACKING

- The results section lacks any mention of searches for the host 18S but it is implied in the methods and discussion. Consider adding a sentence or two about this in the results.

We have now added this information. Based on the premise that the new groundwater symbionts have the same host as the lacustrine *Ca. Azoamicus ciliaticola*, we searched for *Plagiopylea*-related 18S rRNA genes in our metagenomic datasets from which the *Azoamicaceae* genomes were retrieved. We recovered 2 full 18S rRNA sequences from the source data for *Ca. Azosocius aquiferis* and *Ca. Azosocius agrarius*, which have been included in a new 18S rRNA phylogenetic tree. Additionally, a fragment of an 18S rRNA sequence retrieved from the source data for both *Ca. Azoamicus viridis* and *Ca. Azoamicus soli* has also been included in the tree. The tree (new Extended data figure 7) and its description corresponding results and discussion sections of the revised manuscript at lines 384-402 and 464-497.

- Similarly, were attempts to identify ciliate transcripts in the metatranscriptome? Perhaps the 18S is hiding in there or even some obvious eukaryotic transcripts (e.g., actin, tubulin). Ciliates can have a different genetic code, so this is often used to identify ciliate transcripts.

This is a good suggestion, unfortunately, the coverage of the metatranscriptomes was very low (see Supplementary Table S7) and therefore no ciliate transcript recovery was attempted. However, we did recover three 18S rRNA sequences from the metagenomes, which we now report in the manuscript (new Extended data figure 7; see also answer above).

Additionally, we called genes using prodigal (optimized for prokaryotes) with both codon table 11 and 6 on the metagenome assemblies where we recovered the *Azoamicaceae* genomes from. We then compared the resulting amino acid sequences to the ciliate sequences present in the NCBI NR, and indeed found tubulin genes that could be of ciliate origin. Unfortunately, when attempting to follow up on this finding, we realized that molecular data on *Plagiopylea* is extremely scarce, even compared to other protist clades. There is only a single (partial) *Plagiopylean* protein sequence deposited in genbank (AFB18225.1). We are therefore not confident that the available reference data is sufficient to contextualize any coding sequences retrieved from the metagenomes and we hope you can understand our decision not to include this data in the revised manuscript.

- Is there any speculation about what the host might provide to the symbiont? H₂? Are there H₂-uptake hydrogenases still encoded by the symbiont?

There are no hydrogenases encoded in any of the symbiont genomes available to date. The symbiont is likely fully dependent on the host for provision of e.g. most cofactors, nucleotides, amino acids and phospholipids, as no biosynthetic pathways nor many of the transporters are encoded in the symbiont genomes.

However, most importantly, we speculate that the host provides the symbiont with an organic electron donor for denitrification, such as malate or other di-/tri-carboxylic acid. Many protist hydrogenosomes also import malate from the cytosol, which is then converted into pyruvate via malic enzyme. In the symbiont genomes, a putative malate transporter and malate dehydrogenases are encoded, and malate dehydrogenase is also highly expressed in *Ca. Azosocius aquiferis*. We have now rephrased the corresponding sentences on line 319-320 to better reflect this.

Considerations about the host:

- There are dozens of papers out now speculating on the nature of protist:prokaryote symbioses, and indeed in the ciliates it's arguably the most well characterised or speculated upon. So the present manuscript leaves me wanting to know more about what the host is doing (to be fair, this information was sparse in the original paper describing this particular system likely because it's EXTREMELY difficult to retrieve information about the host). But one 'easy'-ish way to test for this would be to look for eukaryotic transcripts in the metatranscriptomic assemblies. The authors discussed the presence of the 18S, but I wonder if there are other transcripts expressed from the host that could link in with the genes being expressed in the bacterium. To do this, you would have to assemble the transcriptomes and do 'tblastn' with candidate genes important for the symbiosis. These could be the hydrogenase genes that are quite unique in the ciliates compared to other eukaryotes (see Boxma et al. 2007 BMC eco evol) or other important genes that have been described in recent studies describing the hydrogenosomal/MRO proteomes of anaerobic ciliates and their relationships to their symbionts (e.g., Lewis et al. MBE 2020; Lind et al. ISME J 2018, Rotterova et al. Current Biology 2020). The ciliates sometime have an alternative genetic code that you can use for deciphering bacterial vs eukaryotic transcripts (run it with codon table =6) . As an aside, I tried to run these BLASTs myself, but the assembly was taking too long...

We agree with the reviewer that more information on the host is key to understanding the nature of this symbiosis. In the original Graf *et al* paper, several transcripts of genes belonging to the putative host were recovered, including eg. Fe-only hydrogenase, pyruvate dehydrogenase, as well as transcripts for six different enzymes of the cytosolic glycolysis pathway for the generation of pyruvate (or malate) (Suppl. Discussion, and Suppl. Table 10, 11 and 12 of that paper).

However, compared to those data, the coverage of the putative Azoamicaceae hosts in the available metatranscriptomes of the Hainich CZE was very low. Therefore, we could only look for ciliate genes in the metagenomes, and unfortunately, this approach did not yield data that we could confidently use for a discussion about the interaction of the host and symbiont (see comment above).

- It seems like the authors could only find 2 different 18S rRNA sequences belonging to ciliates in the different transcriptome projects which makes constructing phylogenetic trees challenging. Were there any attempts made to resolve these 18S sequences in a larger phylogeny of ciliates? It would be interesting to see if the ciliates and the symbionts have co-evolved like has been previously reported in other ciliates (Lind et al. 2018 ISME J) or if there has been 'symbiont swapping' that has been reported as well (Boscaro et al. 2022 Current Biology) and reviewed broadly in (Husnik et al. 2021 Current biology).

As suggested, we have now included a phylogeny of the three newly retrieved Plagiopylean 18S sequences within the broader context of all closely related sequences present in the Silva-NR database (173 reference sequence of Plagiopylea and additional CONThreeP ciliates) (new Extended data figure 7). The phylogeny includes two assembled full 18S rRNA sequences assigned to ciliates found in the metagenomes, from which genomes of *Ca. Azosocius aquiferis* and *Ca. Azosocius agrarius* were retrieved (Hainich aquifer and California groundwater, respectively). In addition, we have attempted more targeted recovery of 18S sequences from the datasets from which *Ca. Azoamicus viridis* and *Ca. Azoamicus soli* were recovered, by specifically assembling reads mapping to CONThreeP 18S rRNA sequences. We successfully managed to obtain a fragment of ~900bp that potentially represents the 18S rRNA gene of the putative ciliate host of *Ca. Azoamicus viridis*. This 18S rRNA fragment is also included in the phylogenetic tree in Extended data figure 7 and all relevant details were added to the methods section at lines 672-682.

We observe that the 18S phylogeny and the Azoamicaceae genome phylogeny appear congruent, which could be indicative of vertical inheritance of the Azoamicaceae, as indicated by the extremely reduced nature of the Azoamicaceae genomes. However, it should be pointed out that apart from the original lacustrine *Ca. A. ciliaticola* no direct link between these recovered 18S rRNA sequences and the respective Azoamicaceae symbionts exists thus far. We have now added this to the discussion in line 464-497.

- In general, a lot of protist literature is overlooked and it could be worth looking at some of the hypothesis pieces that have come out about ciliate symbioses (to complement the insect systems that are currently referenced). For example, Rotterova et al J Euk Micro 2022.

We fully agree and we apologize for this omission. We have now added more contextualization of our findings with regard to protist literature, in the introduction (lines 48-51) as well as to the discussion (lines 457-497 and 505-509).

GENERAL

- Line 45 is redundant to line 42.

We can see how this is confusing. Line 42 refers to respiratory capacity of the ciliate host, whereas line 45 refers to the respiratory capacity of the gammaproteobacterial endosymbiont. In addition to the rephrasing of line 42 discussed above, we have rephrased line 45 (now line 58) from:

“*Ca. A. ciliaticola* lacks the capability for aerobic respiration and is obligately anaerobic,”

to:

“The endosymbiont *Ca. A. ciliaticola* also lacks the capability for aerobic respiration,”

- Line 99 - why is it striking that the genomes are the same size?

We thought it striking that the genomes were almost the same size, since they are clearly distinct by 16S and genome similarity metrics. This in itself may not deserve that much emphasis and therefore we've changed “strikingly” to “very”

- are the cMAGs closed genomes? or linear?

These cMAGs are closed circular genomes. This was previously stated in the abstract, and now also reiterated at the first mention in the results section.

Response to the reviewer comments:

Reviewer #2:

All of my points have been addressed, thank you to the authors for that, and congratulations on your great contribution.

We thank reviewer #2 for their kind words and help in improving this manuscript

Reviewer #3:

The authors have satisfied my concerns and I think addressed many of the concerns of the other reviewers.

We thank reviewer #2 for their supportive review and their help in improving this manuscript

Two comments:

Its interesting to hear that the potential interaction is NOT based on hydrogen exchange (i missed this in my first read through of Graf et al. years ago); I would guess that the MRO has nothing to do with the symbiosis then unless malate is exported from the organelle (possible)? Just out of curiosity, did you ever see the MRO near the symbiont (as has been reported in other ciliate systems)?

We indeed think there is no direct interaction between the MRO and the Azoamicaceae symbionts. Due to the difficulty of enriching the ciliates, Microscopy on these organisms has been extremely challenging. It is currently not known what the localization of the Azoamicaceae symbionts relative to any other cellular components are. Enrichment cultures of the host will be useful in further clarifying this and other questions of cellular integration of these fascinating symbioses.

Just a note for future projects, there are not a lot of protist sequences available on genbank, so its not so surprising that attempts to find ciliate homologues came up short. If you haven't already checked, perhaps the EukProt v3 database could be of use - they have lots of predicted proteomes (blast server) and some transcriptomes (figshare) from ciliates including two Plagiopyla.

Thanks for pointing us towards the EukProt v3 database. We'll be sure to include it in our future work.